# Relevant Information for the Accountability of Private Institutions of Social Solidarity: Results from Fieldwork

Helena Inácio [1,2], Alberto J. Costa [1,2], Ana Maria Bandeira [3], Augusta Ferreira [1,4], Brízida Tomé [3], Carla Joaquim [5], Carlos Santos [1,4], Cristina Góis [6], Denise Curi [1], Deolinda Meira [3], Graça Azevedo [1,4], Mafalda Jesus [7], Maria Goreti Teixeira [7], Patrícia Monteiro [7], Rúben Duarte [1] and Rui Pedro Marques [1,*]

1   Higher Institute of Accounting and Administration, University of Aveiro, 3810-193 Aveiro, Portugal; helena.inacio@ua.pt (H.I.); alberto.costa@ua.pt (A.J.C.); augusta.ferreira@ua.pt (A.F.); carlos.santos@ua.pt (C.S.); denpec11@gmail.com (D.C.); graca.azevedo@ua.pt (G.A.); rubenduarte@ua.pt (R.D.)
2   GOVCOPP—Research Unit on Governance, Competitiveness and Public Policies, University of Aveiro, 3810-193 Aveiro, Portugal
3   Higher Institute of Accounting and Administration of Porto, Polytechnic Institute of Porto, 4200-465 Porto, Portugal; bandeira@iscap.ipp.pt (A.M.B.); brizida.tome@gmail.com (B.T.); meira@iscap.ipp.pt (D.M.)
4   Centre for Research in Accounting and Taxation, Polytechnic Institute of Cávado and Ave, 4750-810 Barcelos, Portugal
5   Department of Economics, Management, Industrial Engineering and Tourism, University of Aveiro, 3810-193 Aveiro, Portugal; carlamfreitas@ua.pt
6   Coimbra Business School Research Centre | ISCAC, Polytechnic Institute of Coimbra, 3045-093 Coimbra, Portugal; crgois@sapo.pt
7   Confederação Nacional das Instituições de Solidariedade, 4050-492 Porto, Portugal; mj.tecnico@cnis.pt (M.J.); maria.goreti.teixeira.1@gmail.com (M.G.T.); pm.tecnico@cnis.pt (P.M.)
*   Correspondence: ruimarques@ua.pt

**Abstract:** The social economy (SE) has emerged as an interesting alternative for dealing with social problems. However, there are some concerns related to the abilities of these institutions regarding accountability. Thus, the present work aimed to determine if private social solidarity institutions (IPSS) are prepared to meet management requirements by increasing their accountability. In Portugal, IPSS are social economy organisations. Using an exploratory focus, we conducted qualitative research on 31 Portuguese IPSS. Interviews with those responsible for these entities took place between June and July 2019. The interviews were guided based on a semistructured script that was created based on a literature review. After content analysis, it was found that, in most of the institutions interviewed, the board does not use management tools, such as performance analysis, social impact assessments, strategic planning, and quality management systems, even though they recognise the importance of using them. This is due to the lack of access or knowledge about their use. In addition, the majority of the IPSS interviewed showed concern about the transparency and ethics of managers. Current strategic management practices are remarkably targeted at companies in the for-profit sector and can compromise the principles of investment in human and social issues.

**Keywords:** accountability; management; nonprofit organisations; social economy; social solidarity institutions; transparency

## 1. Introduction

According to data from the European Union (EU), there are 2 million SE companies in Europe, representing 10% of all of the companies operating in the region. They employ more than 11 million people, i.e., about 6% of jobs (EU 2021). These organisations have the primary objective of serving their members and not obtaining the return on investment in the same way as traditional capital companies do (EU 2021). Social economy organisations

contribute to the accelerations of collective and economic development, which gives them high levels of institutional recognition (Briones Peñalver et al. 2012).

However, in addition to these positive impacts on economic indicators (contribution to the gross domestic product or to total employment), the importance of governance in SE organisations has translated into an increase in the complexity and diversity of social systems (Almeida 2010). SE organisations are, however, intrinsically complex entities due to the diversity of the stakeholders they serve; their distinct organisational structures; their combination of volunteer and paid employees; their dependence on various sources of funding; and the complex issues that they seek to address, which range from alleviating poverty and social exclusion to the promotion of human rights, religious beliefs, and specific ideologies (Speckbacher 2003; Hall and O'Dwyer 2017). This complexity influences the way they are controlled and held accountable for their actions and the impacts of the actions and, consequently, their organisational profile (Hall and O'Dwyer 2017). This sophistication suggests that the entity directly connects with its stakeholders to transmit relevant information in terms of management and strategy.

However, it is essential to note that the management concepts and measurement tools of the for-profit sector are not always transferable to the nonprofit sector (Speckbacher 2003). Many of the decisions that are made in SE sector organisations are made based on nonaccounting information, making the accountability process complex since it is supported by factors that are more difficult to quantify, particularly those concerning the social results of their activities (Connolly and Kelly 2011). Thus, this demonstrates the immediate impact of this type of organisation on the community when decisions are made based on nonaccounting information (directional accountability) (Aimers and Walker 2008), and transforming a series of qualitative information into indicators that are easy to understand for the various stakeholders of these organisations is required.

While market forces and public policies influence the configuration and behaviour of most nonprofit organisations (NPOs), they are generally malleable and flexible in form and require robust internal guidance capabilities to be pointed in the right direction and to ensure success over time (Young 2001). It turns out, however, that in certain situations, such as the death of a founder, the rapid growth of the organisation, or the emergence of new social needs, challenging identity dilemmas can arise, and the resolution of their identities (sometimes new) is crucial for the choice of successful strategies and structures (Young 2001). Therefore, it is recognised that when designing an efficient strategy and when implementing cohesive planning, it is necessary to know the field, numbers, changes, and developments over time.

In the context of Portuguese SE entities, one of the most important groups is represented by IPSS (acronym in Portuguese for private institutions of social solidarity). IPSS are a multisecular reality in Portuguese society and are dispersed throughout the country. However, from the 20th century onward, these institutions maintained or increased their activities, but the State assumed political responsibility for social protection through the consecration of rights and various services (Sousa 2012). IPSS follow and comply with the same legislation as other organisations but are autonomously managed and are not interfered with, with the exception of the supervision of government agencies, namely the Institute of Social Security (ISS).

That said, this work aims to understand whether IPSS are prepared to meet management requirements by increasing their accountability. In this sense, the following research question arises: Are IPSS prepared to meet management requirements by increasing their accountability?

To this end, qualitative research with an exploratory focus was carried out from June to July 2019. Thirty-one IPSS were interviewed during this period. The data were submitted to content analysis, which allowed classification into six categories.

This paper is divided into five sections as follows: after this first introductory sections, the authors present and clarify the theoretical background that supports this work in the second section; the third section presents the methodology; the data analysis and the

findings of the research are presented in the fourth section; and finally, the fifth section presents the final considerations and conclusions.

## 2. Literature Review

### 2.1. Social Economy

The term "social economy" first appeared in France during the first third of the 19th century, and for a long time, its meaning was much broader and amorphous than it is today (Defourny and Develtere 1999). Theoretically, the social economy is closely related to the discipline of economics, and, as a result of its history, to the institutions and the names to which it is linked (Lévesque and Mendell 1999).

Anyone can develop their own a priori conception of the social economy, placing emphasis on economic or social dimensions, or on both. Ultimately, any economic phenomenon that has a social dimension and any social phenomenon that has an economic dimension can be considered to be part of the social economy (Defourny and Develtere 1999).

The basis of the social economy is the creation of self-help mechanisms that are capable of mitigating the social impacts suffered by the less favoured populations. Over the years, it has become possible to identify these mechanisms in almost all countries and throughout all time periods, even before the establishment of the capitalist model. They can be identified in the Egypt of the pharaohs, in ancient Greece, in the Roman Empire, in the dynasties of ancient China, in pre-Columbian America, and in precolonial Africa, among others, and their presence is always identified as a form of mutual aid, including assisting in organising funerals; forming craft guilds; conducting commerce; or meeting people's everyday needs by providing aid, charity, and various other types of assistance (Archambault 1997; Defourny and Develtere 1999).

These mechanisms took many forms and had many names: sororities, guilds, charities, fraternities, merchant associations, trade associations, communities, master associations, guild masters, etc. With the fall of the Roman Empire, monastic associations (convents, monasteries, abbeys, priories, commandos, lodgings, and retreats) gained strength (Defourny and Develtere 1999).

In Portugal, the Misericórdias were founded at the end of the 15th century. The first institution that was created was the Santa Casa de Misericórdia in Lisbon, which sought to carry out 14 works of the Christian mercy cristã (Macías Ruano et al. 2020). The Misericórdias spread throughout Portugal, its colonies, and territories. In the 16th century, the Misericórdias became the main social assistance institution throughout the Portuguese Empire.

In Spain, the Misericórdias were founded a few years later and were inspired by the Portuguese case, but they were always linked to a specific charitable act, unlike the more generic charitable work conducted in Portugal (Macías Ruano et al. 2020).

The Catholic Church played an important role in the history of the social economy, either as an entity responsible for carrying out certain social support activities, as an organiser of these entities (Nitsch 1990), or even as a promoter of political debates that eventually resulted in the training of economists who became major influencers of the social economy, with the objective of reducing the impoverishment of the population (Solari 2007; Moulaert and Ailenei 2005).

In the mid-16th century, the Council of Trent, which established the guidelines for the ecclesiastical control of all welfare brotherhoods as well as established public policies for social assistance in an effort to oppose enlightened despotism and the liberal State, made it very difficult to maintain the private ownership of social assistance facilities (Macías Ruano et al. 2020).

At that time in Europe, a voluntary group could not exist outside the jurisdiction of the Church, State, or some other institutional power unless it had a specific form (Defourny and Develtere 1999).

From the 18th century onwards, the Friendly Societies of England grew in number. In order to provide subsidies to members in the event of illness or death, these institutions charged their members regular fees in order to maintain the entity. This model was adopted in the United States, Australia, and New Zealand. Freedom of association began to make advances in several European countries (England, Germany, and The Netherlands), and especially in the United States. However, it was the workers and peasants' associations of the 19th century, which were inspired by various ideological currents, that had the greatest impact on the evolution of the social economy, from its beginnings to the present day, which can be marked by the emphasis on political and ideological logical pluralism (Defourny and Develtere 1999).

### 2.1.1. The Economic Perspective

In the 1830s, the publication of a treatise on social economy by Charles Dunoyer in Paris introduced the term to economic and academic circles (Dunoyer 1830; Arpinte et al. 2010; Lévesque and Mendell 1999). From then on, economists appropriated the term due to its proximity to the economy, and its shapes and formats adapted to the characteristics of each era or country, especially in European countries (Mudura 2015; Defourny and Develtere 1999).

In his theoretical perspective, Dunoyer (1830) defended the freedom of the economy and the removal of State intervention by the principle of self-help, which ended up influencing the formation of cooperatives, welfare organisations (Arpinte et al. 2010; Solari 2007), and mutual societies (Solari 2007). In his work, he places the importance of the worker above the importance of the State in his various ideas of activity (Dunoyer 1830).

Another economist worth mentioning is Charles Gide. He marked the golden period of the French social economy and materialised the spirit of solidarity through which it was possible to abolish capitalism and the proletariat without sacrificing private property or the freedoms resulting from the revolution (Caeiro 2008). Gide (2016) fostered the ideas of solidarity and cooperation in order to promote a review of the capitalist system and, ultimately, the disappearance of the State, with his ideas considered to be the precursors of the Keynesian movement. Gide proposed a social economy based on mutual aid, modifying the individualistic moral of modern capitalism. Gide's solidarity school proposed a social economy composed of four dimensions of action (Gide 2016):

- Work should consider increasing wages and leisure time and fairness in the relationship between capital and work;
- Services, the objective of which should be personal fulfilment and comfort;
- Social security in order to ensure security in the future and to avoid social risks;
- Economic independence.

It was, however, the sociologist Frédéric Le Play who popularised the concepts of social economy by questioning the passage from "prosperity to the decadence of societies", and later by questioning the living conditions of workers in European families (Le Play 1855). Le Play inaugurated a Society for Social Economy and a journal of the same name (Higgs 1890). Le Play's work, which was based on his international experience as a mining engineer, advocated for the development of cooperatives with the aim of helping workers without, however, promoting a radical transformation of society (Higgs 1890; Kalaora et al. 1989; Caeiro 2008; Le Play 1855; Mendell 2003).

A contemporary of Le Play, Archbishop Wilhelm von Ketteler, was probably more active in addressing major issues of social Christianity—both in terms of theorising and in concrete initiatives to encourage the birth of labour associations (Solari 2007).

It then follows that the social economy goes back to the practices of interclass solidarity (Caeiro 2008) as a way of imposing an ethical vision of society on the individualist and materialist perspective of the liberal political economy (Solari 2007). Its concepts arise as a reaction to the economic and social transformations of the industrial revolution and are influenced by the thoughts of the utopian socialists of the 19th century (Caeiro 2008) and as a form of reaction to the mishaps of the private sector economy and the economy of the State

(Solari 2007). Thus, at its origin, the main concern was with bad employment conditions, with the unemployed, with the less privileged, and with the creation of structures to alleviate the social problems (health, housing, education) experienced by these individuals.

### 2.1.2. Weakening of the Social Economy

Between 1945 and 1975, there was the rise of the welfare state model, which corrected the weaknesses of the capitalist economy (redistribution of income, allocation of resources, and anticyclical policies), mainly in Western Europe. During this period, the social economy lost strength on the European continent. Economic growth in the West consolidated the socioeconomic roles of the public and private sectors, and the social economy played a very limited role between the economy and the State. In Central and Eastern European countries, centrally planned communist economies only allowed state economic activity, and when cooperatives operated, they were stripped of their traditional voluntary and democratic organisation and association (Arpinte et al. 2010; Mendell 2003).

The crises of socialism and the welfare state put the social economy back on the political agenda as an approach that proposed the reconstruction of the relationship between the market, state, and society as a political project, not only with regard to companies or organisations, but also including the individual (Vienney 1994). When the engine of economic growth begins to stutter, the formal distribution mechanisms begin to fail, and new social forces develop and give rise to alternative institutions and mechanisms of solidarity and redistribution as a means of coping with the failures of institutions and of socioeconomic movement to guarantee solidarity between economic agents (Moulaert and Ailenei 2005).

Thus, at the end of the 1970s, the social economy acquired greater significance in terms of economic activity and social policy planning in the European Union countries and in the Americas due to the increase in unemployment. Thus, the social economy returned as an alternative to the need for employment and sources of income, disqualification and social exclusion, the rise of neoliberal ideology and economic policies, and reduction accompanied by state welfare provision (e.g., health, education, certain welfare services). The sum of these needs meant that certain goods and services had to be provided at affordable prices. Thus, the cost to predominantly vulnerable social groups was covered, at various levels, by the social economy sector, or third sector (Arpinte et al. 2010).

### 2.1.3. Resurgence of the Social Economy

The social economy is understood as a set of private entities that are properly organised and endowed with autonomy and freedom of adhesion that are constituted to satisfy the needs of the members who are part of them and the production of goods and services while also ensuring their financing (Catarino 2012). They are not part of the public sector, nor are they controlled by it. They promote democracy and practice activities that allow them to satisfy the needs of individuals and their families (Catarino 2012; Mudura 2015). The value system and principles of popular associations (associationism, associative democracy, cooperativism, and mutualism) served to formulate the modern concept of social economy, which is structured around three large families of organisations: cooperatives, mutual societies, and associations, with the recent addition of foundations and social enterprises (Arpinte et al. 2010).

In this way, the social economy can be understood as "that sphere constituted by companies and organisations that are distinguished by the objective of bringing together an association of individuals instead of shareholders, producing goods and services to satisfy the needs of the members of that association" (Vaillancourt and Lévesque 1996, p. 3).

The social economy has the potential to provide opportunities for people and local communities to participate in all stages of the process of local economic regeneration and job creation, from the identification of basic needs to the operationalisation of initiatives. The sector covers the economic potential and activities of self-help and cooperative movements, that is, initiatives that aim to satisfy the social and economic needs of local communities

and their members. This sector includes cooperatives, self-help projects, credit cooperatives, housing associations, partnerships, community businesses, and businesses.

By nature, the social economy is related to economic activity and is eminently a social activity (Caeiro 2008), so it is said to adopt a market-based organisation to create social value (Miles et al. 2014). Therefore, social economy organisations are hybrid organisations whose main mission is their social purpose, meaning that the concern with financial results stems more from the need to generate resources to sustain their social mission than from the desire to maximise profits (Martínez-Campillo et al. 2018; Miles et al. 2014).

Therefore, the social economy seeks to respond to social needs that are not met by the State (Martínez-Campillo et al. 2018) or the business sector; hence, some authors consider the third sector as a social economy, which often leads to confusion on the topic. The concept of social enterprise in the United States is generally much broader and more focused on companies for the sake of revenue generation than in other definitions (Kerlin 2006).

The exposition of the social economy, which presents itself in a number of diverse ways depending on the country where it is implemented, introduces financial sustainability as a challenge. In Europe, for example, the social economy covers a range of terms used in the various member states, such as "solidarity economy", "third sector", "platform", or even "third system", and activities across Europe that share the same characteristics can be considered to be part of the social economy (Toia 2008).

From a Western European perspective, the social economy is composed of cooperatives, mutual societies, associations, foundations, and other companies and organisations that share the founding characteristics of the social economy (Toia 2008). Over time, the social economy has been a way of overcoming the problems of massive long-term unemployment, social exclusion, well-being in rural and degraded urban areas, health, education, quality of life, population aging, and even sustainable growth (Monzón and Chaves 2008), making it an interesting alternative to deal with current problems.

### 2.1.4. Other Related Concepts
#### Social Entrepreneurship

The conception of social entrepreneurship is the idea of transforming profit maximisation and wealth creation—the ultimate goal of for-profit organisations—into the means by which the "social entrepreneur" satisfies unmet social needs and that the social benefit, which is the ultimate goal of nonprofit organisations, becomes the true "business idea" that needs to be explored, managed, and realised (Arena et al. 2015). While social and economic goals are pursued by these organisations, they are market-oriented but mission-focused (Wolf and Mair 2019).

#### Hybrid Companies

For some social enterprises, activities aimed at serving beneficiaries (and therefore at the pursuit of social objectives) are separate from those aimed at serving customers, thus generating revenue; for other organisations, they are the same. The latter have been referred to as integrated hybrid organisations and are particularly prevalent in the European context. In the US, it is more common to follow a differentiated approach, where business activities are viewed as a source of income and viewed and kept separate from the organisations' social objectives (Wolf and Mair 2019).

#### Nongovernmental Organisations (NGOs)

NGOs are elusive accounting entities that really seem to defy definition (Gray et al. 2008). The term NGO or nongovernmental organisation was introduced in 1945 due to the need for the UN to differentiate the participation rights of specialised, intergovernmental agencies and international private organisations in its statutes. A more modern definition defines NGOs as:

> Any voluntary, non-profit, citizen-oriented group at the local, national or international level. Task-oriented and led by people with a common interest, NGOs perform a variety of humanitarian services and functions, bring citizens' concerns to governments, monitor policies and encourage political participation at the community level. They provide analysis and knowledge, serve as early warning mechanisms, and help monitor and implement international agreements. Some are organised around specific issues such as human rights, the environment or health. (UN 2021)

Hudson and Bielefeld (1997) specified that NGOs are organisations that:

- Provide useful goods or services (in some specified legal sense), thus serving a specified public purpose;
- Cannot distribute profits to persons in their individual capacities;
- Are voluntary in the sense that they are created, maintained, and completed based on voluntary decisions and initiatives by members or a board;
- Exhibit value rationality, which is often based on strong ideological components.

They may have a global hierarchical structure, with a strong central authority, or a more flexible federal arrangement. They may be based in a single country and operate transnationally. In addition, they have a structure that is independent of government control, cannot be linked to political parties, cannot be profit-oriented, and cannot belong to a criminal group since NGOs must be nonviolent institutions (Willetts 2002).

Nonprofit Organisations

In the United States, the term "nonprofit organisations" prevails. In that country, the concept of social enterprise is broader and is more focused on the company in favour of revenue generation than it is in other definitions (Kerlin 2006). It is noteworthy, however, that there is a divergence of understanding between academics and professionals in the area: for academics, social enterprise is based on a continuum that ranges from profit-oriented companies that are engaged in socially beneficial activities through a hybrid purpose (with economic and social purpose) to those with a fully social purpose (even with commercial activities that support this mission). The business community, on the other hand, understands social enterprises as being nonprofit (Kerlin 2006). In Canada, the term "social economy" is predominant, and in Latin America, the terms "third sector organisation" or "social enterprise" are used (Moulaert and Ailenei 2005; Mendell 2003).

*2.2. The IPSS in the Context of the Social Economy*

In Portugal, the concept of social economy is a concept that is legally anchored in the activity developed, in the legal form of the entity that develops it, and in the principles that guide its operation. It should be noted that the concept of social economy is a European concept and that there are differences in the delimitation of what is considered third sector in the Anglo-Saxon approach and what is considered to be part of the social economy in the continental European approach, the former referring to its activity and the (non)pursuit of profit, that is, including nonprofit organisations and charities, while the second focuses on the type of entities that are legally demarcated based on three parameters: activity/purpose, legal form, and guiding principles. However, the social economy, third sector, and nonprofit sector are expressions that are widely used synonymously (Meira 2013; Fajardo García 2012; Almeida 2010).

Law No. 30/2013, implemented on 8 May, which approved the Basic Law of the Social Economy (LBES), defines the social economy (ES) in paragraph 1 and in paragraph 2 of article 2 as "the set of economic and social activities, freely carried out by the entities referred to in article 4 [ . . . ]" and as activities that "are intended to pursue the general interest of society, either directly or through the pursuit of the interests of its members, users and beneficiaries, when socially relevant" (DRE 2013).

From this law, it follows that the legislator associates the notion of SE with a specific social object, which can be translated into the exercise of an economic and social activity, the purpose of which is the pursuit of general interest.

According to Namorado (2006), the expression "economic activity" refers to an activity related to the production of goods and services, under the aegis of a rationality that implies the maximisation of results, cost containment, and reproducibility of productive virtualities. It is an activity developed according to business criteria, that is, according to criteria of rationality and economic sustainability (Fajardo García 2012).

Zurera (2011) emphasises the idea that despite the relevance of the social dimension, these entities base their activity on profitability criteria, which are subject to the cost–benefit discipline. In this way, we are dealing with entities that develop activities that are aimed at achieving a lower cost of goods or services for the benefit of the members or the community than what would be obtained by other means.

However, this includes social activity as well as economic activity. In this way, the intention of the legislator is to highlight that the activity carried out by HE entities does not have a profit purpose, but that it instead has the purpose of satisfying the needs of both members and the community through their participation in said activity (as is the case of mutual societies) (Fajardo García 2012). Thus, it is understood that one of the guiding principles of higher education is the primacy of the person and of social objectives over capital (Meira 2013).

As for the purpose pursued—general interest—it is related not only to the fact that these entities pursue social purposes, emerging as partners of the welfare state and co-operating with it to guarantee that citizens have the vital minimum of economic, social, and cultural rights ("a public–social partnership" assumed between the State and HE entities), but also to its peculiar mode of organisation and functioning, which is distinct from the public and private sector and is reflected in its guiding principles, among which the conciliation between the interests of its members, users, or beneficiaries and general interest stands out. However, regarding the pursuit of this general interest, the LBES admits that it can be pursued directly or indirectly through the promotion of the interests of the members, users, and beneficiaries (Meira 2013).

Entities included in the Statute of Private Institution of Social Solidarity (EIPSS), a statute that includes legal persons without profit purpose (DRE 2014), can be used as examples of entities that directly pursue purposes of general interest:

> constituted exclusively on the initiative of individuals, with the purpose of giving organised expression to the moral duty of justice and solidarity, contributing to the realization of the social rights of citizens, provided they are not administered by the State or by another public body. (DRE 2014, article 1)

Therefore, we are dealing with entities whose main objective is social solidarity and who have a clear mission to support situations of economic and social vulnerability based on an assistentialist paradigm of social intervention (Almeida 2011).

The terms of article 4 of the LBES incorporate the following entities into the notion of the social economy, provided that they are incorporated in national territory (DRE 2013): (a) cooperatives; (b) mutual associations; (c) mercies; (d) foundations; (e) private social solidarity institutions (IPSS) not covered by the preceding paragraphs; (f) associations with altruistic purposes that act in the cultural, recreational, sports, and local development spheres; (g) entities covered by the community and self-managed subsectors that are integrated under the terms of the constitution in the cooperative and social sector; and (h) other entities with legal personality that respect the guiding principles of the social economy, provided for in article 5 of this law and that are included in the social economy database. This list shows that we are dealing with private entities with a legal personality and not individuals or public entities.

Furthermore, the LBES does not adopt the legal form of entities as an exclusive criterion for subjective delimitation. Effectively, the legislator in addition to the legal forms corresponding to the traditional delimitation of social economy families (cooperatives,

mutual societies, associations, and foundations) also addresses a legal statute (the EIPSS), contained in Decree-Law No. 119/83 from 25 February (DRE 1983), which was profoundly amended by Decree-Law No. 172-A/2014 from 14 November.

Article 5 of the LBES sets out the guiding principles that must be observed by social economy entities in the exercise of their activity and consequently in their organisation and functioning (Meira 2017). The aforementioned rule enshrines that social economy entities are autonomous and act within the scope of their activities in accordance with the following guiding principles (DRE 2013):

- The primacy of the person and social objectives;
- Free and voluntary membership and participation;
- Democratic control of the respective bodies by their members;
- Conciliation between the interests of members, users, or beneficiaries and general interest;
- Respect for the values of solidarity, equality and nondiscrimination, social cohesion, justice and equity, transparency, shared individual and social responsibility, and subsidiarity;
- Autonomous and independent management of public authorities and any other entities outside the social economy;
- The allocation of surpluses to pursue the purposes of social economy entities in accordance with general interest without prejudice to respect the specificity of the distribution of the surpluses, an action that is inherent to the nature and substrate of each social economy entity constitutionally.

### 2.3. Strategic Management of SE Organisations

One of the central concepts in the study of business management concerns the strategy and purpose of an organisation (Bartlett and Ghoshal 1994).

An organisational strategy is the set of policies and practices to address the purposes of the organisation (Young 2001). The strategy cannot be confused with the entity's purpose, which is unique and demonstrates the reason for the organisation's existence. The purpose should translate what it hopes to achieve and all of the activities that must be developed (Drucker 1973; Hitt et al. 2012). Usually, the purpose of an SE organisation is not to produce profit but to focus on a social objective, be it poverty reduction, community development, sustainability, or health and social assistance (Hall and O'Dwyer 2017).

The differences between profitable companies and those belonging to the TSO are sufficient to suggest that NPOs may need their own strategy formulation concept, which differs from for-profit companies. For Moore (2000), among these differences, two are more remarkable than the others: (a) NPOs define the value they deliver in terms of the organisation's mission and not according to their financial performance, and (b) they often guarantee their revenue through people who (voluntarily or unintentionally) pay for benefits for people other than themselves, unlike a customer who buys goods for their own benefit.

Mission statements describe how an organisation achieves its purpose. These are high-level statements and generally aim to provide a clear and succinct summary of why the organisation exists (Hitt et al. 2012). The mission defines the organisation's value to society and creates the organisation's objective, so in some cases, it becomes the metric used to judge past performance and evaluate future courses of action (Bryce 2017). However, it cannot be seen as an action plan subject to periodic change. On the contrary, the mission statement must be a contract, a contractual promise, the terms of which are secure until the contract is renegotiated and approved by its component parties (Bryce 2017). A company is not defined by its name, statute, or social contract, but rather by its business mission (Drucker 1973). Business objectives are only possible when they are clearly and realistically defined within the organisation's mission and purpose.

In general, the missions of NPOs are defined in substantive rather than financial terms. Thus, when considering NPOs, a "mission" has the following properties (Bryce 2017):

- Social contract: Points to specific public problems that the company seeks to alleviate or the desirable social conditions that the company seeks to achieve (Moore 2000). Failure to perform this task can lead to termination of status, denial of status, and penalties for management and the organisation.
- Permanence: It is permanent unless changed by amendments that are subject to the approval of the members, trustees, and the public, who are represented by the State.
- Clarity: A mission statement is always short (with less than 100 words) and clear, pointing to a specific public service.
- Approval: A mission must be approved by the directors and trustees and accepted by the State in which the NPO is established.
- Proof: It is necessary for NPOs to prove their existence, performance, and mission fulfilment annually as well as for them to report on their use of revenue, expenses, and other resources in order to proceed with the mission.

In addition, a mission is typically targeted at a specific group of clients or beneficiaries, such as a local community of individuals with particular social or health needs, meaning that effectiveness and impact measures involve the assessment of how the organisation has impacted these groups (Hall and O'Dwyer 2017; Werther and Berman 2001); it also becomes necessary to ascertain why the organisation has embraced this cause (Werther and Berman 2001). In this way, the mission must contain at least three elements: for whom the entity works, what the entity delivers, and how the entity delivers its mission.

The strategy adopted by an institution depends on developing a clear mission for the organisation and the definition of its strategic objectives. The mission should be transformed into detailed support objectives that drive the entire organisation (Hitt et al. 2012; Drucker 1973). They should identify the "what" and the "why" of their social responses.

The strategic vision should seek to balance the interests of the different stakeholders to ensure the continuity of participation for each one. While the mission is the reason for the organisation's existence, the visions represent the desired generic purposes for which the company's efforts are guided (Werther and Berman 2001). In the formation stage of an NPO, the vision usually reflects a notion of grand purpose (glory) but evolves to embrace critical constituents. As the organisation succeeds and matures, the vision can be expanded to a larger purpose or domain (Werther and Berman 2001).

This expansion typically reflects the organisation's growing success and pressures placed on internal constituents (such as managers, professionals, and staff) and external forces (such as funding sources, clients, board members, community expectations). While the founder or those who are involved in the training stage may see this expansion of the vision as desirable, they may also see it as a dilution of the organisation's original mission (Werther and Berman 2001). The vision booster is usually the founder or founding group. As maturity increases, the vision becomes increasingly institutionalised and driven by employees and supporters and less by the founders (Werther and Berman 2001).

As for the objectives and challenges of organisations, it is suggested that they be established at an early stage and in a very spontaneous way, such as through brainstorming. Nevertheless, basic ideas are vital to hierarchise the objectives and challenges within certain classifications through debates. Oliveira (2010) suggests four levels for the objective hierarchy process: mission, company objectives, functional objectives, and challenges.

Thus, based on the assumptions of Hitt et al. (2012), the strategy must start from the development of a clear mission and its strategic objectives, and, in combination with the fact that the vision is related to the maturity of the institution (Werther and Berman 2001), it is advocated that the existence of a mission, vision, and strategic objectives may indicate that the institution has a strategic plan. Moreover, it has been justified that the maturity of the mission may be related to the presence of the following elements: for whom the entity works, what the entity delivers, and how the entity delivers the mission (Werther and Berman 2001).

For a strategic approach that allows us to see an organisation from a broader perspective, covering both its leadership and its constraints, some questions are needed, such as (Werther and Berman 2001):

1. How is the nonprofit trying to achieve its goals?
2. What are the expectations of those who support the organisation?
3. What strategies are available to the organisation?
4. What roles do the leaders play?
5. What resources does the organisation have to support its goals?

Thus, one can understand how successful NPOs achieve success. However, it is necessary to consider that "best practices" are just one of several factors that determine how NPOs achieve their goals (Werther and Berman 2001).

However, organisations need to know who they are by defining their organisational identity (Young 2001). The organisational profile should provide an overview of the organisational characteristics by reporting primary activities and identifying how these activities relate to the organisation's mission and key strategic objectives (e.g., poverty reduction, environment, human rights) (GRI 4 2013).

Organisational identity can be defined as what is central, distinct, and lasting about an organisation (Albert and Whetten 1985), and it is also related to the organisational role or function and can even be described in terms of what an organisation does or "what business it is in" (Young 2001). Behind their identity, there must be a certain flexibility that allows NPOs to choose whom they want to be among several possible organisations (Young 2001). Identity is a distinct but holistic concept that integrates, supports, and drives several operational and management concepts that guide the direction and character of a long-term organisation (Young 2001).

SE organisations occupy a unique space within the economy, where, as companies, they are driven by the need to be financially sustainable, but similar to NPOs, they use economic surpluses to boost social and environmental growth (Zainon et al. 2014). In this context, it should be emphasised that financial crises may call the identity of an NPO into question (Young 2001). To resolve any financial deficits, these organisations often find ways to supplement their funding sources (governmental or philanthropic) through activities that complement their main activity. However, it should be noted that this "complement" should have the sole purpose of fulfilling the social mission of the organisation.

In this way, we perceive that organisational identity is an important condition for realising whether the institution is prepared to meet the management requirements in the rendering of accounts.

### 2.4. Strategic Planning

Faced with a dynamic and complex operational environment, NPOs are increasingly attentive to their organisational sustainability (Al-Tabbaa 2012; Claeyé and Jackson 2012) in the long term. To meet the needs of the external environment, strategies are needed to enable these organisations to remain effective and achieve their purposes (Moore 2000). In this sense, the personnel, governance, and financial requirements of a social purpose organisation follow the organisation's identity (Young 2001).

Strategic planning will help the board of directors reflect on the social interests associated with the mission and the skills required to fulfil it. The staff need to support the purposes, experience, and sensitivity of the organisation to the social problems being addressed; be aware of the business problems faced by the organisation; engage volunteers with the cause; and assess the necessary funding needs, which should include additional government donations as well as charitable funds to the same causes (Allison and Kaye 2015; Bryson 2018).

The use of strategic management tools can help SE organisations find better balance and ensure their long-term sustainability. They should focus on the impact of sustainability trends, risks, and opportunities on the organisation's long-term perspectives and financial performance. Strategic planning will provide the paths, courses, and action programmes

that must be followed in place to achieve the objectives and challenges set by the organisation (GRI 4 2013). However, as current strategic management practices are notably oriented towards companies in the lucrative sector, they can compromise the principle of investments in human and social issues (Kong 2008).

*2.5. Quality Management System*

One of the management tools that has shown a growing interest among SE organisations is quality management systems (QMSs) and excellence (Al-Tabbaa et al. 2013, Cairns et al. 2005). This interest may be justified by increased competition between these organisations to obtain scarce resources; intensifying pressure from funders, government agencies, and other stakeholders to provide services effectively; and higher user needs for providers to deliver value for money.

QMS models see performance improvement as an approach to combat this difficult environment (Cairns et al. 2005; Kong 2008). Total quality management (TQM) is suggested as a more holistic approach to relevant performance improvement, as it encompasses social and technical issues (Bou-Llusar et al. 2005).

Thus, developing institutional tools to help NPOs work on their identity-related issues is of paramount importance. Thus, their definition and articulation should be the first responsibility of senior management (Bartlett and Ghoshal 1994). In this work, it is assumed that the function manual is an integral part of the QMS, but it is admitted that, in some situations, organisations have a function manual without having a QMS. The same reasoning was adopted concerning global performance assessment (Kong 2008).

*2.6. Governance*

Governance issues are relevant to all organisations but are of particular concern to NGOs concerning the "values" to which they aspire but also in terms of resource management and performance (GRI 4 2013). According to Almeida (2010), the greater centrality of the third sector in governance can be measured through its contribution to the market for goods and services, participation in the labour market, and the consequent dynamisation of local economies.

The current concern with organisational/corporate governance can be attributed to the (Spear 2004):

1.  "Excessive executive power that can culminate in abuse of pension funds, substantial remuneration packages for executives, corrupt practices, as well as poor decision-making;
2.  A concern that systems that try to allow owners to exercise control over managers have often been ineffective and complex;
3.  A concern that, with the increase in the globalisation of corporations and the relatively weak regulatory powers of national governments, some effective restriction on the power of corporate managers is necessary;
4.  A growing concern for the environment and the failing market for common property (the tragedy of the commons); and thus, put more significant needs on a good administration."

The purpose is the centrepiece of corporate governance in the SE, and its purpose is to provide a set of rules, principles, and institutions that make their processes more efficient while generating value for their relevant stakeholders at the same time (Speckbacher 2008). By learning more about the reasons for possible inefficiencies in NPOs, one can discuss which governance mechanisms help facilitate cooperation and have improved engagement among stakeholders (Speckbacher 2008).

Organisational governance is the result of the composition and behaviour of the board of directors and how they deal with stakeholder expectations. Governance performance is a specific behaviour domain. Good governance practises are expected to positively impact organisational decision making, positively influencing the organisation's other performance areas (Crucke and Decramer 2016). However, it has been observed that

governance practices targeted at for-profit organisations can be detrimental to the mission of SE organisations (Wolf and Mair 2019).

The different stakeholders expect NGO decision-makers to ensure that their organisation reflects the diversity of the society in which they operate and to act with equity and integrity in their leadership and management (GRI 4 2013). The board must bring together an ethical framework for all aspects of governance to pursue its objective effectively and correctly (Hitt et al. 2012). On the other hand, governance deficiencies can damage the organisation's reputation and fundraising capacity and can make it difficult for the organisation to achieve its objectives (Rassart and Miller 2013).

Therefore, NPOs should not only be well governed but should also be seen as being well governed to (Rassart and Miller 2013):

- Reduce funding from traditional sources such as governments, corporations, and private donors;
- Compete with other NPOs facing similar funding difficulties;
- Increase the demand for services resulting from reductions or cuts in programmes by governments;
- Manage more complex and sophisticated entities, as many NPOs have grown in size and complexity;
- Hold greater responsibility and expectations for an increasing number of stakeholders who may have conflicted expectations for the organisation;
- Ensure the rapid dissemination of information across social media, which can quickly affect the way the organisation is viewed;
- Overcome difficulties in recruiting quality board members who may choose not to join the organisation's board due to time constraints or liability concerns.

Oversights in financial management concerning governance are seen as being especially important by NPO stakeholders given that many NPOs rely on donors or public support. In addition to the responsibility to comply with laws and regulations, it is clear that there is a need for high and balanced accountability measures not only for donors but also for other critical stakeholders who are affected, the community, and society at large (GRI 4 2013).

In NPOs, the social bodies are primarily volunteers. However, their responsibilities—and associated liabilities—are generally not (Rassart and Miller 2013). The remuneration of the corporate bodies reflects the incorporation of ethical aspects. A balance between the responsibilities assumed and the perceived remuneration is expected.

Although women have historically played essential roles in charity work (Themudo 2009), there are fewer women holding executive director and board member positions when compared to the number of men in those positions (Pynes 2000), and pay gaps are also evident (although not as pronounced as in the lucrative sector) (Lewis et al. 2013). The strength of an NPO is directly related to female empowerment and the relative position of women in society (Themudo 2009). Women generally reveal stronger preferences for equity and collective good than men, and the higher society's demand for social products and the lower the participation of governments are, the more women enter this area (Themudo 2009). However, few scientific studies address this issue since it is understood that a framework to evaluate SE organisations should contemplate the parity between women and men in social bodies.

The discussion about worker participation in the management bodies is related to the agency's theory or even to the view of the shareholder (Friedman 1970) versus the view of the stakeholder (Freeman 1999). The shareholder's view emphasises shareholder value as the company's goal, in contrast to the stakeholders' perspective, which argues that an equally legitimate goal is to serve the interests of employees and other stakeholders. Thus, while employee participation is a logical phenomenon from the point of view of stakeholders, from the shareholders' point of view, it is only considered desirable when it serves the interests of shareholders (Kleinknecht 2015).

In SE organisations, the founders' interests are oriented towards social causes, which should be aligned with the interests of employees. In addition, for an organisation to effectively and appropriately pursue its purpose, the people who are involved in the organisation must share a common understanding of how this is carried out (Campbell and Alexander 1997; Ireland and Hitt 1999). For that purpose, an organisation will have to have values that express what the organisation considers suitable and the principles that express what the organisation considers correct (Hitt et al. 2012).

Together with the purpose, these values and principles form the ethical structure of an organisation. This structure guides people's decisions and should be reflected in their policies, systems, and processes (Hitt et al. 2012). Since councils are mechanisms for employee participation in (or at least a critical analysis of) decisions at the organisational level, it seems sensible to say that it is essential to assess whether workers are involved in governing bodies (Kleinknecht 2015).

Corporate governance practices are primarily determined nationally through (different) legal norms (Porta et al. 1998).

### 2.7. Transparency

Finally, institutional legitimacy (verification that the organisation has respected its "rules", status, mission, action programme) and the legal rules that are applicable to its legal form should be communicated to its stakeholders (Bagnoli and Megali 2011). The transparency of the governance process and its relationship with the mission and vision are seen as of particular importance by the main stakeholders of the NGO (GRI 4 2013).

The purpose of an NPO is public benefit, but general terms and benefits are defined in various ways, and as a result of this, there are specific regulations, laws, reports, and exemptions from taxes and fees (Pointer and Orlikoff 2002). In addition, the activities that are carried out by these organisations cover a wide range of stakeholders, including donors of funds, both public and private; project and activity beneficiaries; people related to those beneficiaries; volunteers; and workers. All of these groups require information about the activities carried out by the organisations with which they have some relationship (Santos et al. 2018).

These organisations have a moral obligation to meet the informational needs of these groups; in short, they have an obligation to be transparent. NPOs should transmit the added value provided by their activities to society, implementing the actions they consider necessary to do so, resulting in greater transparency (Cabedo et al. 2018).

### 3. Research Methodology

This paper aims to answer the following research question: Are IPSS prepared to meet management requirements by increasing their accountability?

To substantiate the fieldwork, a semistructured interview script was established, which was created based on a previous bibliographic review. The script used for the interviews is in Appendix C. In qualitative studies, fieldwork is often used to engage with the phenomenon to gather information/data or to analyse in situ practices (Appendix D).

To be successful, however, fieldwork must be preceded by prior planning where the object and purpose of the research are clarified (Feldman 2019), who will be investigated and what will be investigated are determined, and how they will be investigated is decided, which implies the structuring of a roadmap of questions (Jacob and Furgerson 2012). These steps, which have been added to the definitions of the respondents' approaches, comprise the research protocol presented in Appendix C.

### 3.1. Sample Definition and Data Collection

It was not possible to conduct fieldwork with all entities of the IPSS population (5358) at the time when the study was conducted. Thus, a representative sample of the study population was defined by following appropriate statistical procedures. Thus, to define the number of IPSS to be included in the sample, Epi Info software version 7.1.0.6 (Dean

et al. 1991) was used by adopting a 90% confidence level and a margin of error of 5%, which resulted in a total of 258 IPSS being interviewed. This number still did not seem feasible, so the margin of error was increased to 10%, meaning that only 67 IPSS had to be interviewed. However, depending on the availability of the respondents, which was affected by the vacation period of the institutions, only 31 visits were made. The interviews were scheduled in the month of May 2019, and the visits took place in June and July 2019.

Several other questions were raised: the legal nature, the type of social responses, the dimension of the IPSS, and the geographical area, and more areas were added later on.

### 3.2. Methodology for Data Analysis

To analyse the reports prepared in relation to the data collected during the field work, the content analysis methodology was adopted (Bardin 2004). There are two content analysis methods (Henry and Moscovici 1968): (i) closed procedures, in which the categories are predefined prior to the analysis itself, and (ii) open or exploratory procedures, where there are no predefined categories. Throughout this project, the closed procedure was adopted since the categories were predefined in the literature review phase and are presented in Table 1.

**Table 1.** Identification of categories.

| Units of Record (URs) | Categories |
|---|---|
| UR1—Identity of the investigated organisation<br>UR2—Primary activities and support activities<br>UR3—Values<br>UR4—Transformations<br>UR5—Number of users | C1—Organisational Identity |
| UR6—Mission<br>UR7—Vision<br>UR8—Strategic objectives<br>UR9—Strategic planning realisation | C2—Organisational Strategy |
| UR10—Quality management system<br>UR11—Performance analysis<br>UR12—Function manual | C3—SGQ |
| UR13—Number of employees in management positions<br>UR14—Number of women in management positions<br>UR15—Remuneration of managers | C4—Governance |
| UR16—Transparency | C5—Transparency |

The interviews were digitised, and the fieldwork data were analysed. The content analysis methodology was adopted (Bardin 2004) with the help of the program NVivo12 Version 12.6.0 (QSR International, Melbourne, Australia). This software was chosen because it has the possibility of encoding and categorising various data formats, minimising investigator bias.

### 4. Data Analysis

During the content analysis process, the main ideas gathered from the interviews were identified and grouped into five stages based on similarity. First, the respondents were coded. To maintain data confidentiality, the names of the IPSS have been replaced by the acronym IPSS and numbered from 1 to 31. IPSS27 is a Holy House of Mercy, IPSS14 is a cooperative, IPSS12 and IPSS18 are parish social centres, and the others are associations. In this way, all of the different IPSS types are represented in this work. The analysis units, or units of record (URs), were identified in the second stage. In the third step, the URs were grouped into categories according to Table 1.

In the fourth stage, the categories were analysed, and, in the fifth stage, they were interpreted.

### 4.1. C1—Organisational Identity

All of the entities that were surveyed have a clear view of their main activities and are oriented to this end. As for the support activities, it was identified that four IPSS carry out support activities for the Social Security Institute (ISS) but do not receive any remuneration. Some of the elderly homes (IPSS8, IPSS9) and educational institutions (IPSS1, IPSS7, IPSS8) that were investigated have part of their activities supported by a particular (for-profit) activity. In addition, the IPSS reported performing instrumental activities, such as social stores (IPSS7, IPSS26), canteens (IPSS13, IPSS21), sale of products for recycling (IPSS13), space rental for events (IPSS7), and paid tours (IPSS6, IPSS13, IPSS26). These activities guarantee additional money that can then be reverted to the institution.

As for the profile of the organisations that were interviewed, it was identified that:

- Many have their origins in certain religions:
    - IPSS1, the Anglican Church;
    - IPSS6, IPSS11, IPSS12, IPSS17, IPSS18, IPSS25, the Catholic Church;
    - IPSS23, the Evangelical Church.
- Some are rooted in community solidarity:
    - IPSS6, IPSS9, IPSS10, IPSS14, IPSS15, IPSS19, IPSS25, IPSS26, IPSS27, IPSS28, IPSS29, IPSS30, IPSS31.
- Two were started by parents and are linked to mental and physical health:
    - IPSS2 and IPSS3.
- Entities that provide activities for the elderly, such as day centres, have some historical relationship with groups of workers:
    - IPSS5 and IPSS13.

Another point to note is that many institutions that started their activities focusing on childhood have extended their actions to the care of the elderly (day centres, nursing homes, home care services) (IPSS7, IPSS8, IPSS30) and vice versa (IPSS5, IPSS14).

IPSS4, IPSS6, IPSS14, IPSS15, IPSS18, IPSS19, IPSS20, IPSS22, IPSS29, and IPSS31 do not have clearly defined values, in the other IPSS, the following words showed high rates of repetition:

- Respect (14);
- Solidarity (13);
- Responsibility (7);
- Cooperation (6);
- Ethics (6);
- Dignity (5);
- Transparency (5);
- Equality (5);
- Confidence (4);
- Quality (3);
- Christian(s) (3).

### 4.2. C2—Organisational Strategy

Following legal proceedings, the IPSS that were visited have a clearly defined mission that is available on their official website or on the Facebook page of the institution. The exception is for IPSS21 and IPSS26 that do not have a mission statement. The interviewees knew of the existence of the mission, but they could not state its terms. The investigation team analysed the mission statements of the other 29 entities interviewed and realised that only the mission statements for IPSS22, IPSS25, and IPSS29 did not contain the three fundamental criteria: what they do, how they do it, and for whom they do it.

Of the 31 IPSS investigated, 21 (67.7%) had a clearly defined vision, i.e., a future perspective for the institution. For four IPSS, it is possible to improve the vision, and six (19.4%) do not have a clearly defined vision. Despite this result, not all of the surveyed organisations are aware of the real importance/meaning of having a vision outlined.

Among the 31 entities interviewed, only 26% carry out strategic planning. IPSS31 does not do so in a systematic manner, IPSS12 claims the need to improve the process, and IPSS23 is in the implementation phase.

Among those that do not carry out strategic planning (74%), IPSS12 realises the need to do so, IPSS13 states that the president "does it in his head", IPSS15 states that it only makes the budget due to obligation, IPSS16 only does it on-demand, IPSS21 "lives the day-to-day", IPSS does not have one but considers it interesting, and IPSS31 says that they "used to have one, but with the reduction of staff and users, it no longer does it".

### 4.3. C3—Quality Management Systems

Only two of the IPSS that were investigated have a quality management system. IPSS8 is certified by ISO 9001/2015 and is transversal to all responses, and IPSS12 has the Equass European Quality in Social Service program. Among the 31 IPSS investigated, only 12 have an institutionalised performance evaluation system, and IPSS14 replied that:

> "Yes, but this past year it was not done. This year it went wrong because we wanted to get our model right. It was because we do not think it was fair. After all, we do the group evaluation and self-assessment, and often self-assessment puts the average up. We want to change it in order not to give too much focus to self-assessment. The weight of self-assessment is too high. The staff react well. There was a year where they did not react well because someone said they did not like a person, and it did not go well." (IPSS14)

IPSS10 claimed to have an informal evaluation system, IPSS18 does not have a performance evaluation system, and IPSS1 is in the course of implementing one; the respondents from IPSS2, IPSS25, and IPSS31 regret the fact that they do not have one. IPSS9 stopped doing it and commented that "it can never be done well". According to IPSS21:

> "The official way, we do not have it. The board is talking, getting to know the problems of the clients. But it has been carried out informally. Perhaps it is implemented with the new management. There is no employee's performance evaluation. I am not aware that a form of evaluation is defined because we cannot confront an employee without a system of rules. We do not argue to evaluate. There has to be a set of predefined goals. The new direction is discussing this."

None of the IPSS investigated claimed to have a function manual identifying tasks, responsibilities, and autonomies.

### 4.4. C4—Governance

When asked about the values present for entity management, 10 (32.2%) of the IPSS did not have one or did not answer the question. Among those who answered, 33.3% mentioned the word ethics in their organisational values.

As for the presence of women on the management boards, contrary to what happens in the operational areas, where the presence of women is predominant, it was observed that men are the majority in the studied IPSS. Furthermore, even when women were in management positions, they held less prominent positions such as substitutes or secretaries of the bureau. Women were chairmen of the board in only 10 of the 31 entities interviewed: IPSS2, IPSS4, IPSS7, IPSS8, IPSS14, IPSS16, IPSS18, IPSS23, IPSS25, and IPSS28.

However, it should be noted that in parish social centres, the presidency is usually exercised by a priest of the Catholic Church.

Concerning the wage issue, Portuguese legislation imposes some limitations on the payment of salaries of directors of social bodies, as noted in the theoretical framework. However, the chairman of the board of IPSS12 is remunerated and exercises the role

of chairman of the board in five other IPSS belonging to the Catholic Church. It was not possible to identify whether nonfinancial benefits are offered to the occupants of management positions.

Finally, regarding the participation of workers and nonmembers in the social bodies, Portuguese legislation once again imposes restrictions, so we were unable to find workers in the social bodies. It is worth mentioning, however, that representatives of the governing bodies were identified as performing operational tasks. This situation was observed in IPSS6, IPSS7, IPSS9, IPSS15, IPSS16, IPSS21, IPSS25, IPSS26, IPSS28, IPSS29, IPSS30, and IPSS31.

*4.5. C5—Transparency*

When asked if the existence of a microsite can increase the transparency of an IPSS, out of the 31 IPSS interviewed, 20 answered yes. They believe that having a website increases the transparency of the institution.

IPSS1 believes that "transparency and social impact are imperative to 'sell' IPSS and reach other spheres (enterprise-level) and the community". IPSS3, IPSS11, IPSS13, IPSS23, and IPSS27 stated that they are already transparent in disclosing their information.

IPSS21 trusts that "we have to seek transparency", and IPSS8 suggests that transparency "can put institutions on an equal footing" and that it can facilitate the exchange of information between institutions. "In a company, we can do it, in the social area this is more difficult."

IPSS26 suggested that IPSS are not transparent: "Will everyone be transparent in disseminating information?", "Today, we know it is not like that."

Furthermore, IPSS31 stated that "there are things we cannot put on the site. Ideas, projects, things that are in our thoughts. We have some antibodies in the village that we must neutralise."

## 5. Final Considerations

The goal of this paper was to ascertain whether IPSS are prepared to meet management requirements by increasing their accountability. According to Bartlett and Ghoshal (1994), the central concepts in the study of business management include understanding the strategy and purpose of the organisation. According to Young (2001), the central concepts in the study of business management include understanding the strategy and purpose of the organisation. That said, this study sought to investigate the policies and practices adopted by the investigated institutions through semistructured interviews conducted with the leaders of the institutions. From the answer, the units of record (URs) were extracted, which allowed five categories to be created: C1, Organisational Identity; C2, Organisational Strategy; C3, Quality Management System; C4, Governance; and C5, Transparency.

With regard to organisational identity, it was determined that the institutions have knowledge of their social importance, dominate the activities carried out, and often depend on an additional effort to ensure the financial sustainability of their activities. Not only commercial (or exchange) activities were reported, but also the involvement of some managers in conducting operational activities in the day-to-day of the institution in order to save on expenses through the hiring of personnel was reported, a fact that was clearly mentioned during some of the interviews. In most of the interviews, the influence of the organisations' founders was noticeable, which goes against the ideas of Young (2001) and Werther and Berman (2001), which suggest that over the years, the identity of the institution can distance itself from the vision of its founders. On the contrary, there was a strong presence of the ideas and social ambitions of the founders of these institutions.

In the operational strategy category, with the exception of educational institutions, UR analysis identified that there are no concerns about increased competition between IPSS, as suggested by Al-Tabbaa et al. (2013) and Cairns et al. (2005). On the contrary, the need to form networks to complement knowledge and needs became apparent. Most of the entities interviewed do not have structured strategic planning, even recognising that

the realisation of a planning process may be necessary for the institution. It was observed that the managers have a long-term view for the organisation; however, managers are sometimes far removed, and they do not realise that there is competition or other issues related to strategic planning. The missions analysed in the present study included the three components suggested by the literature (what, who, and how); however, most of the respondents did not know the mission even though all of them worked to fulfil the institution's purpose. Moore (2000) defends the importance of strategic planning to ensure its effectiveness and the achievement of an organisation's purposes.

Concerning category C3, it was perceived that the institutions recognised the importance of having a QMS, agreeing with the arguments of Kong (2008) and Bou-Llusar et al. (2005), which draw attention to the fact that QMSs can contribute to improving organisational performance. Despite this, it was noticed that an incipient number of institutions have a QMS and a function manual, while the percentage of organisations with a performance manual was slightly higher. It is interesting to note the response of two of the institutions investigated:

> "No. We are still in development, but it is a very complex area. Performance evaluation can be a conflict area. This may be a pandora's box, but we create injustices both when we do it and when we don't." (IPSS5)

> "I have tried because I think it is important. Nevertheless, we do not. I am good at evaluating, but without a methodology, people end up feeling bad." (IPSS29)

Concerning the governance of entities, the interviewees' discourse was clearly in favour of the company's management being transparent regarding their ethics. Only one of the presidents interviewed was in a paid position, as he reported being both the president and an employee of the institution, making the situation a little nebulous since he received a salary as an employee. In one of the institutions interviewed, the respondent stated that the president preferred to keep donated paintings in his residence "to prevent them from being damaged". Concerning gender equity, it was observed that (i) operational positions are notably occupied by women, and the justification for this is because women prefer to be cared for by women; (ii) the presence of women in management positions is much smaller than that of men, around 15–20%, even when considering that women in management positions such as presidency correspond to 30%; and (iii) in religious institutions (parish social centres), the presidency positions are mainly occupied by priests, which contributes to a reduction in the number of women in CEO positions.

All of the institutions reported the importance of transparency, C5, and understand that ethics and transparency only strengthen the institution's reputation, contributing to increased funds and the attraction of new donors and users.

Thus, in summary, we can say that institutions are aware of their importance and activities and maintain the vision of their founders over time. Institutions are worried about day-to-day activities and fail in structured strategic planning, failing, for example, to identify competitors or partners to complement their needs. Although institutions recognise the importance of a QMS, few have implemented one, with many also failing to implement performance evaluation. It should also be mentioned that there is still no gender equity in terms of both operational and management positions. We also found that everyone recognises the importance of ethics and transparency.

There are two limitations that have been identified in the present research. The first concerns the period in which the interviews were conducted: close to the vacation period. The second concerns the workload of the respondents, who were not always available to talk to the research team.

This work is expected to contribute to the research gap related to social indicators, specifically nonfinancial indicators, and will hopefully assist SE organisations in their management process and accountability.

**Author Contributions:** Conceptualisation, A.F., G.A., A.J.C. and R.P.M.; methodology, D.C., B.T. and C.J.; software, C.S., R.P.M., and R.D.; validation, G.A., D.C., A.J.C., B.T., C.J. and R.P.M.; formal

analysis, G.A., D.C., A.J.C., B.T., C.J. and R.P.M.; investigation, G.A., D.C., A.J.C., A.M.B., A.F., B.T., C.J., C.S., C.G., D.M., H.I., M.J., M.G.T., P.M., R.D. and R.P.M.; resources, A.M.B., A.F., C.S., C.G., D.M., H.I., M.J., M.G.T., P.M. and R.D.; data curation, A.M.B., A.F., C.S., C.G., D.M., H.I., M.J., M.G.T. and P.M.; writing—original draft preparation, D.C. and B.T.; writing—review and editing, G.A., A.J.C., C.J. and R.P.M.; visualization, G.A., A.J.C., C.J. and R.P.M.; project administration, A.F. and R.P.M.; funding acquisition, A.F. All authors have read and agreed to the published version of the manuscript.

**Funding:** The European Regional Development Fund (FEDER), through Operational Competitiveness and Internationalisation Program (COMPETE 2020—POCI), and the Foundation for Science and Technology (FCT) financed this research, with reference number POCI-01-0145-FEDER-030074.

**Institutional Review Board Statement:** Not applicable.

**Informed Consent Statement:** Not applicable.

**Data Availability Statement:** Not applicable.

**Conflicts of Interest:** The authors declare no conflict of interest. The funders had no role in the design of the study; in the collection, analyses, or interpretation of data; in the writing of the manuscript; or in the decision to publish the results.

## Appendix A. Terminology Employed

For the literature review presented in this paper, consider the terms "third sector organisation (TSO)", "nonprofit organisation (NPO)", "nongovernmental organisation (NGO)", and "social economy (SE) organisation" as synonyms. It is important to note here that NGOs are accountably elusive, which seems to challenge the definition of the SE (Gray et al. 2008). The term NGO was introduced in 1945 due to the need for the United Nations (UN) organisation to differentiate in its status rights for specialised agencies, intergovernmental organisations, and international private organisations to participate. A more modern definition describes an NGO as a nonprofit, voluntary citizen's group that is organised on a local, national, or international level to address issues in support of the public good. Task-oriented and made up of people with a common interest, NGOs perform a variety of services and humanitarian functions, bring the concerns of citizens to governments, monitor policy and program implementation, and encourage the participation of civil society stakeholders at the community level (UN 2021).

## Appendix B. Accountability

Accountability is a broad and comprehensive concept that refers, among other things, to taking responsibility for decisions that have been made and explaining and justifying them (Edwards and Hulme 1995; Gray et al. 2008; Peters 1983; Bovens 2016; Santos et al. 2018). Scrutiny and doubts about the public role of NPOs a few decades ago have required better accountability on the part of these institutions (Kearns 1994). Accountability has mainly focused on internal control and auditing, monitoring, evaluation, and compliance with standards and regulations (Choudhury and Ahmed 2002). According to Edwards and Hulme (1995), the only way NPOs can prevent corruption is by developing systems to monitor their performance, accountability, and strategic planning to ensure that a line between transparent commitment and blind preference remains drawn.

In recent years there has been a trend for NPOs around the world to take accountability initiatives and activities that go beyond minimum legal requirements, thereby increasing transparency and good governance, leading to greater trust and reputation (Becker 2018). In this context, accountability initiatives beyond the legal minimum have gained substance in the last two decades (Becker 2018). Thus, depending on the legal regulations of each country, a variety of different codes of conduct, self-regulation systems, certifications, and accreditations have evolved as tools to support good governance in entities in the SE sector.

NPOs need even more management than business companies because they lack the discipline of the final outcome (Speckbacher 2003).

**Appendix C**

Semistructured interview script for visiting entities

Even though the aim of the interview is to generate answers to the questions below, the interview should be conducted in a relaxed manner. Asking generic questions allows us to obtain answers to the questions below.

1. Characterisation:
· The social mission of the institution;
· The activities developed in the institution;
· The number of users covered by the activities;
· The evolution of the institution;
· The major transformations in the activity, if any, and what led to these transformations (a form of institution consolidation: enlargement/retraction).
2. Internal management models:
· Human resources in management positions;
· Composition of management positions;
· Decision-making processes;
· Performance evaluation;
· The existence of a quality management system;
· Strategic planning implementation;
· The respondent's perception of the importance of transparency.

**Appendix D**

Protocol for fieldwork

1. Initial contact was made to schedule the meeting by telephone.
2. The teams were trained by two researchers.
3. An e-mail was sent to the directors of the institutions confirming the appointment.
4. A quick internet search was conducted to verify if the institution had a website.
5. At the beginning of the meeting with the entity, the group adopted the following procedure:
· Thank the institution for their time;
· Present the project and emphasis two points:
  ○ Talk about the constitution of the project team;
  ○ Present the project summary and mention its objectives.
6. At the end of the interview, the group of researchers:
· Demonstrated the website to the entity and collected suggestions;
· Sought opinions on the usefulness of the project;
· Investigated the availability of the project and whether or not it is better to have a website, and whether the organisation would be interested in having a website;
· Thanked the organisation for their time and indicated the next steps;
· Committed to keeping the entity informed on the development of the project.

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
