# Peer review of "Relevant Information for the Accountability of Private Institutions of Social Solidarity: Results from Fieldwork"

_economies, doi:10.3390/economies10020035_

Round 1

Reviewer 1 Report

This is an interesting work, it's a starting point for the improvement of the management of a type of entity of the social economy in Portugal, and it's extensible to any other entity where the ownership of the legal structure is not clearly determined.

Note some reflections to the author:

The first sentence of the article is as drastic as it is inaccurate. The social economy does not emerge as an alternative to face the problems that humanity faces and that are not solved by the State. The origin of the Social Economy begins, in the scientific literature, with Charles Dunoyer, a liberal economist who defends a moral approach to economics, not a State model assuming humanitarian care services, and the development of the content of the so-called Social Economy in Europe ran parallel to the cooperative movement, whose legal manifestation, cooperative societies, being mutual-based societies. They seek to intervene in the market and compete with capitalist business structures, doing the same but in a different way, and always from the private sector, not public sphere. Moreover, manifestations of private initiatives to address human problems have occurred even before the very existence of nation-states. Apart from the private care institutions that have been taking place since the Roman Empire, in Portugal, the first manifestations of their own social philanthropy occurred with the so-called albergarias (inns or guesthouses) located on the pilgrimage routes, usually old Roman roads, during the 11th century (Russell-Wood, "The Santa Casa da Misericórdia in Portugal", in "Fidalgos and Philanthropists. The Santa Casa da Misericórdia of Bahia, 1550-1955", University of California Press, 1968, p. 8).

On the other hand, when the term and content of Social Economy was created, the Portuguese IPSS, in their legislative scope, did not have their own substantive materiality. It is true that there were the so-called Irmandades da Misercórdia, Associações de solidariedade social, de succoros mútuos; o Fundações de solidariedade social, but, as a whole, they did not have a common consideration as IPSS until the Statute das Instituições Privadas de Solidariedade Social (1979), although they had already been partially contemplated in the Administrative Code of 1940, Law No. 1998 (19444) and Law No. 2120 (1963). The paragraph of the article between lines 60 to 67 should be reformulated.

For the correct assessment of the contributions made in the article, it might be convenient to carry out an analysis of the study differentiating the answers and analyzing them within the heterogeneity of the IPSS entities interviewed. Thus, for example, in a process of accountability of the different structures are not the same, nor do the responses that each one of them give the same value, since Cooperatives, Foundations or the Santa Casa da Misericórdia have different purposes, they are accountable to different groups and have different direction and managements (democratic for the owners of the Cooperatives, imposed by the founder in the Foundations, and the hierarchical canonical —or secular character that is discussed— of the Misericordias.

It would also be interesting if in the analysis of the responses these were weighted according to the nature of each of the IPSS that have been interviewed. The relevance of the responses of a Santa Casa da Misericórdia, which has been serving for more than five centuries, and that of a Volunteer Association, which may have a purely temporary nature, must be different.

Recommendations for reading and contributions from authors such as:

  • Almeida, V. (2010): "Governação, Instituições e Terceiro Sector: As Insittuiões Particulares de Solidariedade Social", Tesis doctoral. Repositório científico da Universidade de Coimbra, disponible en https://estudogeral.sib.uc.pt/handle/10316/13315,

  • Macías Ruano, A. J., Pires Manso, J. R., de Pablo Valenciano, J., & Marruecos Rumí, M. E. (2020). The Misericórdias as social economy entities in Portugal and Spain. Religions11(4), 200, disponible en https://www.mdpi.com/2077-1444/11/4/200.

  • Briones Peñalver, A.J., López Castelao, M.P. y Cardoso de Sousa, F. (2012): “La Economía Social Ibérica: el caso de las Santas Casas de la Misericordia de Portugal como Instituciones Particulares de Solidaridad Social”, REVESCO, nº 107, pp. 35-57.

Author Response

We would like to start by thanking you for your comments and contributions to the article, which were generally accepted, and which we believe contribute to the improvement of our article. Anyway we will try to answer your questions individually.

  1. The first sentence of the article is as drastic as it is inaccurate. The social economy does not emerge as an alternative to face the problems that humanity faces and that are not solved by the State. The origin of the Social Economy begins, in the scientific literature, with Charles Dunoyer, a liberal economist who defends a moral approach to economics, not a State model assuming humanitarian care services, and the development of the content of the so-called Social Economy in Europe ran parallel to the cooperative movement, whose legal manifestation, cooperative societies, being mutual-based societies. They seek to intervene in the market and compete with capitalist business structures, doing the same but in a different way, and always from the private sector, not public sphere. Moreover, manifestations of private initiatives to address human problems have occurred even before the very existence of nation-states. Apart from the private care institutions that have been taking place since the Roman Empire, in Portugal, the first manifestations of their own social philanthropy occurred with the so-called albergarias (inns or guesthouses) located on the pilgrimage routes, usually old Roman roads, during the 11th century (Russell-Wood, "The Santa Casa da Misericórdia in Portugal", in "Fidalgos and Philanthropists. The Santa Casa da Misericórdia of Bahia, 1550-1955", University of California Press, 1968, p. 8).

After your comment, we agree that in fact that sentence is not very correct, so you suggestion was taken into consideration, and we changed the sentence in both the abstract and the introduction.

In order to clarify the origin and evolution of social economy entities, we chose to add more content to the literature review to complement this information, since this was also the suggestion of another reviewer.

  1. On the other hand, when the term and content of Social Economy was created, the Portuguese IPSS, in their legislative scope, did not have their own substantive materiality. It is true that there were the so-called Irmandades da Misercórdia, Associações de solidariedade social, de succoros mútuos; o Fundações de solidariedade social, but, as a whole, they did not have a common consideration as IPSS until the Statute das Instituições Privadas de Solidariedade Social (1979), although they had already been partially contemplated in the Administrative Code of 1940, Law No. 1998 (19444) and Law No. 2120 (1963). The paragraph of the article between lines 60 to 67 should be reformulated.

In consideration of your comment, we added some content to the literature review in order support and explain with more detail this paragraph.

  1. For the correct assessment of the contributions made in the article, it might be convenient to carry out an analysis of the study differentiating the answers and analyzing them within the heterogeneity of the IPSS entities interviewed. Thus, for example, in a process of accountability of the different structures are not the same, nor do the responses that each one of them give the same value, since Cooperatives, Foundations or the Santa Casa da Misericórdia have different purposes, they are accountable to different groups and have different direction and managements (democratic for the owners of the Cooperatives, imposed by the founder in the Foundations, and the hierarchical canonical —or secular character that is discussed— of the Misericordias.

As suggested, for better perception, it was specified in the first paragraph of section 4.

  1. It would also be interesting if in the analysis of the responses these were weighted according to the nature of each of the IPSS that have been interviewed. The relevance of the responses of a Santa Casa da Misericórdia, which has been serving for more than five centuries, and that of a Volunteer Association, which may have a purely temporary nature, must be different.

In the first paragraph of section 4, we have specified the number of interviewed entities by nature.

  1. Recommendations for reading and contributions from authors such as:

Almeida, V. (2010): "Governação, Instituições e Terceiro Sector: As Instituições Particulares de Solidariedade Social", Tesis doctoral. Repositório científico da Universidade de Coimbra, disponible en https://estudogeral.sib.uc.pt/handle/10316/13315,

Macías Ruano, A. J., Pires Manso, J. R., de Pablo Valenciano, J., & Marruecos Rumí, M. E. (2020). The Misericórdias as social economy entities in Portugal and Spain. Religions, 11(4), 200, disponible en https://www.mdpi.com/2077-1444/11/4/200.

Briones Peñalver, A. J., López Castelao, M. P. y Cardoso de Sousa, F. (2012): “La Economía Social Ibérica: el caso de las Santas Casas de la Misericordia de Portugal como Instituciones Particulares de Solidaridad Social”, REVESCO, nº 107, pp. 35-57.

The recommendation to include new authors was taken into consideration and have been included in the article.

Thank you very much for your review, which, along with the comments of other reviews, has greatly contributed to the improvement of this article.

Reviewer 2 Report

Dear author,

I believe that the proposed manuscript can be of great value in generating knowledge and that it is a great contribution.

However, there are several points I would like to highlight:

Regarding the scientific literature review, I have doubts about the categorisation of legal entities used. There is talk of NPOs, however, I think that the reality of this type of entity, which straddles the boundary between the company and the association, requires greater specificity. Are they non-profit entities? If so, there is literature concerning social enterprise (EMES Network) that has been neglected. There are authors such as Laville, Pérez de Mendiguren, Defourny, Nyssens, Coraggio, Etxezarreta, Guridi, Monzón y Chaves, Arcos-Alonso, Morandeira and others who can give a broader perspective on the type of enterprise we are talking about. It would be necessary to clearly delimit what type of company we are talking about. Are they immersed in an ecosystem of social enterprises? Are they solidarity enterprises? Social enterprises? What legal entity do they share?
On the other hand, it is a very positive point, in my opinion, to propose a qualitative methodology using programmes adapted to this type of analysis, such as NVIVO 12. However, I believe that more specific information is needed about the categorisation of the variables of analysis: Why one category and not another? How are they related? A single hypothesis does not seem to be sufficient...
The citations need to be adapted to the journal format.
Finally, I miss a clear and concise conclusion section in the manuscript. What are the findings of the research? It should be a brief but clear section that contributes to knowledge.
All this leads me to propose a detailed revision of the article.

Author Response

We would like to start by thanking you for your comments and contributions to the article, which were generally accepted, and which we believe contribute to the improvement of our article. Anyway we will try to answer your questions individually.

  1. Regarding the scientific literature review, I have doubts about the categorisation of legal entities used. There is talk of NPOs, however, I think that the reality of this type of entity, which straddles the boundary between the company and the association, requires greater specificity. Are they non-profit entities? If so, there is literature concerning social enterprise (EMES Network) that has been neglected. There are authors such as Laville, Pérez de Mendiguren, Defourny, Nyssens, Coraggio, Etxezarreta, Guridi, Monzón y Chaves, Arcos-Alonso, Morandeira and others who can give a broader perspective on the type of enterprise we are talking about. It would be necessary to clearly delimit what type of company we are talking about. Are they immersed in an ecosystem of social enterprises? Are they solidarity enterprises? Social enterprises? What legal entity do they share?

Your comment is extremely important and therefore, in the literature review, we made a more detailed clarification on the different terminology adopted in the context of Social Economy.

  1. On the other hand, it is a very positive point, in my opinion, to propose a qualitative methodology using programmes adapted to this type of analysis, such as NVIVO 12. However, I believe that more specific information is needed about the categorisation of the variables of analysis: Why one category and not another? How are they related? A single hypothesis does not seem to be sufficient...

For a better understanding of the origin and classification of the categories, a paragraph was added in section 3.2, which states that these categories were supported in the literature review, included in this version of the article.

  1. The citations need to be adapted to the journal format.

The citations were adjusted to the specificities of the journal.

  1. Finally, I miss a clear and concise conclusion section in the manuscript. What are the findings of the research? It should be a brief but clear section that contributes to knowledge.

We have added a paragraph in the concluding remarks section, summarizing the main conclusions reached.

Thank you very much for your review, which, along with the comments of other reviews, has greatly contributed to the improvement of this article.

Reviewer 3 Report

The present manuscript provides a field experiment on the accountability of private institutions of social solidarity.

  1. Abstract should be improved, within the limits of 200 words, including more literature background.
  2. The research design should be completely rebuilt
  3.  Literature review section: 
    1. sub-sections Accountability and Terminology employed should be eliminated and transformed into an appendix.
    2. This section should be completely re-written
  4. Research Methodology: this section should be completely re-written
  5. Appendices should be better improved

Author Response

We would like to start by thanking you for your comments and contributions to the article, which were generally accepted, and which we believe contribute to the improvement of our article. Anyway we will try to answer your questions individually.

  1. Abstract should be improved, within the limits of 200 words, including more literature background.

Taking into account the comments and in order not to exceed 200 words, the abstract has been adjusted.

  1. The research design should be completely rebuilt

The research design has been adjusted to make it clearer and more understandable.

  1. Literature review section: 
  • sub-sections Accountabilityand Terminology employed should be eliminated and transformed into an appendix.
  • This section should be completely re-written

This section was all rewritten as requested and the sub-sections Accountability and Terminology employed were added to the appendices. 

  1. Research Methodology: this section should be completely re-written

The methodology was adjusted taking into account the comments and suggestions received.

  1. Appendices should be better improved

The Appendices have been adjusted according to the suggestions.

Thank you very much for your review, which, along with the comments of other reviews, has greatly contributed to the improvement of this article.

Round 2

Reviewer 2 Report

Dear authors,
I believe that the manuscript has been greatly improved.
It now meets the necessary conditions for publication.
I would like to congratulate you once again on the work you have done.

Author Response

Thank you for your comments.

The English was also revised.

Reviewer 3 Report

The present manuscript is well-written and the topic is interesting

  • The present manuscript has the necessity a moderate English changes
  •  

Author Response

Thank you for your comments.

The English was revised as recommended.